# Association between the *PPARGC1A* Gly482Ser polymorphism and muscle fitness in Chinese schoolchildren

Qi Wei[ID][1,2]*

**1** Key Laboratory of General Administration of Sport of China, Beijing, China, **2** Hubei Institute of Sports Science, Hubei, China

* 48750028@qq.com

## Abstract

**Data Availability Statement:** Data cannot be shared publicly because all the paticitants are children. Data are available from the Ethics Committees of Hubei Sports Science Insititute (contact via public Email: hbtykx@126.com) for

### Objectives

Muscle health is essential for children's physical development and future health. *PPARGC1A* gene encode the peroxisome proliferator-activated receptor coactivator 1α which coactivates transcription factors that control mediating skeletal muscle fiber type conversion and skeletal muscle fiber formation. The *PPARGC1A rs8192678 Gly/Ser* (Gly482-Ser) polymorphism was associated with the regulation of skeletal muscle fibre type. This paper aims to explore the association between the PPARGC1A rs8192678 (Gly482Ser) polymorphism and muscle fitness in Chinese schoolchildren.

### Methods

We detected the distribution of the *PPARGC1A rs8192678* (Gly482Ser) polymorphism by DNA typing of saliva samples from untrained Southern Chinese Han children aged 7–12 years. Considering that muscle studies in children cannot use invasive sampling, we analyzed the association between alleles and genotypes with high validity tests of muscle fitness assesment in children(handgrip strength, standing long jump, sit-ups and push-ups).

### Results

The results showed no significant differences in height, weight or body mass index between the sexes. The grip strength indicators were correlated with age in boys and height and weight in girls. Sit-ups were significantly higher in girls with the *PPARGC1A* Gly/Gly genotype than in boys, and handgrip strength and standing long jump were significantly lower in girls with the *PPARGC1A rs8192678* (Gly482Ser) genotype than in boys. Genetic model analysis showed that the Gly482 allele had a dominant genetic effect on the Gly482 allele is hypothesized to influence the expression of type I fibers in skeletal muscle in girls, while the Ser482 allele affects on type II fibers in girls. The two alleles had little genetic effect on boys.

researchers who meet the criteria for access to confidential data.

**Funding:** This paper was funded by Natural Science Foundation of Hubei Province (ZRY1869). The funders had no role in study design, data collection and analysis, decision to publish, or preparation of the manuscript.

**Competing interests:** The authors have declared that no competing interests exist.

**Abbreviations:** HGS, handgrip strength; SLJ, standing long jump; BMI, body mass index; VO$_2$max, maximal oxygen consumption; PGC-1α, peroxisome proliferator-activated receptor gamma coactivator 1α; SNP, single nucleotide polymorphism; PCR, polymerase chain reaction; MAF, minor allele frequency; CHS, southern Han Chinese; CHB, Han Chinese in Beijing; KO, knockout; LDL, low-density lipoproteins; MEF2, myocyte enhancer factor 2; MEF2C, coactivator of myocyte enhancer factor 2; CREB1, cAMP-responsive element-binding protein 1; CRTC2, CREB-regulated transcription coactivator 2.

## Conclusions

The results suggested the possible association of the *PPARGC1A rs8192678* (Gly482Ser) polymorphism on myofibril type-related phenotypes in Han Chinese children in southern China, with a particular impact on girls.

## 1. Introduction

Muscle fitness is an integral part of children's physical fitness, reflecting multidimensionally the ability of a muscle or group of muscles to be fast(muscle power/strength), repetitive(muscle endurance) and resistant to fatigue [1, 2]. Smith et al systematic reviewed the relationship between muscle strength and various health indicators in children and adolescents from six external databases and conducted a Meta-analysis showing a high correlation between muscle strength and cardiovascular health, skeletal health, self-esteem index and athletic performance [3]. and is a valuable indicator for monitoring health in children [4] and adolescents [5]. Muscle health in childhood and adolescence was significantly associated with cardiovascular disease [6], unfavourable glucose homeostasis [7], Body mass index(BMI), skinfold thickness, insulin resistance, triglycerides, bone mineral density [8] and premature death from all causes later in life [9].

Convenient and valuable indicators assessing muscular strength in children and adolescents include handgrip strength (HGS) (upper limb strength) and standing long jump (SLJ) (lower limb strength), while tests of muscular endurance are sit-ups and push-ups [10, 11]. HGS is considered the most reliable test for measuring maximum isometric strength in arm muscles [12]. SLJ is an assessed proxy of lower limb muscle strength [13, 14]. The sit-ups test is an indicator to evaluate the dynamic endurance of abdominal muscles [15]. The push-ups test usually assesses upper body muscular endurance [16, 17]. More studies have strongly proven that the muscle strength of children and adolescents (particularly muscle strength) has declined globally over the past few years [10, 11, 15, 16, 18]. Muscle strengthening activities have been added to physical activity guidelines for children and adolescents in Europe and the United States [19–21].

There are three types of human skeletal muscle fibers types: type I, type IIa and type IIx. Wade et al. showed a negative correlation between the proportion of type I muscle fibers and the percentage of body fat. Forty percent of the variation in body fat percentage could be explained by muscle fiber type composition. A low ratio of type I fibers is regarded as a risk factor for inducing obesity and insulin resistance [22]. The ratio of type II fibers is associated with blood pressure. Skeletal muscle content shows prominent interindividual variability in fiber composition: type I fiber from 15% to 85%, type IIa fiber from 5% to 77% and type IIx fiber from 0 to 44% [23]. This variability seems to be the result of genetic factors (heritability estimates roughly range between 45% and 99.5%), age 23 and exercise training as external stimuli [24].

Since the development of sports genetics, research on several genetic polymorphisms from athletes to the general population are associated with muscle fiber composition, particularly the *PPARGC1A rs8192678 (Gly482Ser)* polymorphisms involved in energy metabolic function, cytoskeletal function, or circulatory function [25].

Peroxisome proliferator-activated receptor coactivator 1α (PGC-1α, encoded by the *PPARGC1A* gene located on chromosome 4p15.2) is an important regulator of mitochondrial biogenesis and is involved in the gene expression of key enzymes for fatty acid oxidation and

oxidative phosphorylation. The *PPARGC1A rs8192678 (Gly482Ser)* polymorphism have Gly482 and Ser482 alleles identified to three genotypes(Gly/Gly,Gly/Ser,Ser/Ser). This polymorphism is associated with exercise performance [26]. It is critical in training-induced muscle adaptation as it coactivates transcription factors that control a myriad of biological responses, mediating skeletal muscle fiber type conversion and skeletal muscle fiber formation [27].

In a meta-analysis on variants of the *PPARGC1A* rs8192678 (Gly482Ser) polymorphism, the Gly482 allele and the Gly/Gly genotype were considered beneficial for exercise performance in different types of exercise [28]. The Gly482Ser polymorphism may be detrimental to the exercise-induced conversion of fast to slow muscle fibers. Studies have shown that the Gly482Ser polymorphism affects on muscle fiber type in Japanese nonathletic females, and it is hypothesized that polymorphisms at this allele could facilitate muscle fiber type conversion [29].

These results show that the Gly482Ser polymorphism is associated with muscle fiber type and transformation in athletes and adults. However, there are unrelated reports available regarding possible associations between the Gly482Ser genotypes and the skeletal muscle of schoolchildren. Since the study was conducted on schoolchildren and that invasive sampling of muscle types was not possible, we chose a muscle phenotype-related indicators as the most valid tests for muscle fitness assessment. This study was designed to explore the distribution characteristics of the *PPARGC1A* rs8192678 (Gly482Ser) polymorphism in untrained schoolchildren aged 7–12 years, and also tested four tests (HGS, SLJ, sit-ups and push-ups) as skeletal muscle fitness indicators to evaluate the association.

## 2. Methods

Schoolchildren were recruited for this study from a primary school in the urban area of Yichang City, Hubei Province, and were randomly selected from schoolchildren aged 7–12 years old, with the following selection requirements: 1) be physically fit, free of serious illness, no history of mental illness and no professional sports initiation or training 2) Consent from parents and signed a written informed consent form. The final subjects were 200 boys and 150 girls. The study was performed in accordance to the Helsinki Declaration regarding the conduct of clinical research and was approved by the Ethics Committees of Hubei Sports Science Institute (ECHSS-2020001).

The saliva samples were collected using the saliva collector Oragene DNA OG-500 collection tubes (DNA Genotek, Canada). Then genomic DNA extraction was performed from the saliva sample with the PUREGENEs DNA purification kit (Gentra Systems, Minneapolis, MN, USA). The genotyping of the *PPARAGC1A* Gly482Ser (rs8192678) polymorphism was conducted using the F-5′ TAAAGATGTCTCCTCTGATT 3′ and R-5′ GGAGACA CATTGAACAATGAATAGGATTG 3′ primers. The polymerase chain reaction (PCR) cycling conditions were carried out by TaqMan SNP (Applied Biosystems, USA) and the ABI PRISM 7500 Sequence, with an denaturing step for 5 min at 94˚C; 40 cycles for 45 s at 95˚C, annealing for 30 s at 64˚C 30 s and extension for 1 min at 60˚C. The analyzed PCR product was digested by HpaII (ThermoFisher Scientific, Massachusetts, USA) and sequence analysis was performed to determine the genotype [26, 29].

Subjects were measured in the school gymnasium and required in light clothing in the morning, and trained technicians measured weight and height, BMI was estimated using the equation BMI = weight/height$^2$ (kg/m$^2$).

HGS: HGS was measured by a Jamar Hard dynameter(Sammons Preston, Canada). Knowing which hand is the dominant hand before the test, the peak force recorded by the

dynamometer is used to represent the maximum hand HGS for each subject. Both hands were tested three times, whichever was the highest value. During the test, the subject was required to stand upright with the feet separated at shoulder width and the elbows fully extended; then the grip was squeezed with full force and held for at least two seconds and was recorded in kilograms [12].

(2) Standing Long Jump: The subject stands behind the starting line, jumps forward as far as possible and lands with feet together and remains upright. The distance from the starting line to the heel was recorded [13, 14]. Repeat the test twice and keep the best score in cm.

(3) Sit-up: The participant lies relaxed on the mat, crosses the arms over the chest and bends the knees at 90 degrees. An assistant pressed the foot in preparation for the test. On the start signal, the participant rises to a sitting position with his or her elbows straightened so that they touch the knees, then quickly lies down. The correctly performed number of times the forehead touched the knee was recorded in 30 seconds [15].

(4) Push-ups referenced the DMT 6–18 to assess upper body strength in schoolchildren with a good validity [16]. Students start by lying face down on the mat, and both hands are placed on their backs. The exercise starts with the hands under the shoulders in a push-up (without bending the knees), then touching one hand to the top of the second hand and again placing the hands under the shoulders, returning to the resting position under control. A push-up movement is counted as completed when both hands touch the back again. The tester demonstrates the correct movement that the student must perform the correct movement at least twice before performing the test. The test lasted 40 seconds. The number of repetitions of the correct movement was recorded.

Statistical data is expressed as the means ± SDs, using the SPSS 22.0 software. The genotype and allele frequencies of the *PPARGC1A* rs8192678 (Gly482Ser) polymorphism were tested with Hardy-Weinberg equilibrium (HWE). The association between Gly482Ser and skeletal muscle fitness (HGS, SLJ, sit-ups, push-ups) was assessed by multivariate linear regression analysis with three genetic models (additive, dominant and recessive). The significance level was set at $p < 0.05$.

## 3. Results

The *PPARGC1A* rs8192678 (Gly482Ser) polymorphisms were within the Hardy–Weinberg equilibrium (p = 0.946). The main characteristics of the study participants are presented in Table 1.

As presented in Table 1, there was no difference between the sexes in height, weight or BMI, with a significant difference between the boy and girl groups in the standing long jump

**Table 1. Characteristics and genotypes contribution of the participants.**

|  | All (n = 350) | Boys (n = 200) | Girls (n = 150) |
|---|---|---|---|
| Age (years) | 10.06±1.23 | 10.45±1 | 9.63±1.32 |
| Height (cm) | 142.79±9.63 | 144.42±7.73 | 140.97±11.24 |
| Weight (kg) | 38.88±11.83 | 40.23±10.38 | 37.36±13.27 |
| BMI (kg/m$^2$) | 19±3.89 | 19.2±3.46 | 18.75±4.43 |
| HGS (kg) | 16.8±6.19 | 17.18±3.49 | 16.31±4.97 |
| Sit-ups (number) | 22.18±6.20 | 21.68±4.92 | 22.81±7.56 |
| SLJ (cm) | 140±20.13 | 145.44±20.67** | 133.15 ±17.47 |
| Push-ups (number) | 16.08±4.35 | 16.06±4.41 | 16.11±4.36 |

* presents the significant difference in SLJ between sexes.

**Table 2. Associations of *PPARGC1A* rs8192678 G/A (Gly482Ser) polymorphisms with HGS, SLJ, sit-ups and push-ups in boys and girls.**

| Gene Name (rs Number) | | Genotype | | HWE p-value | | p-value | |
|---|---|---|---|---|---|---|---|
| *PPARGC1A* (rs8192678) | | | | | Additive | Dominant | Recessive |
| ALL | GG (30%, n = 96) | GA (40%, n = 162) | AA (30%, n = 92) | 0.16 | GG vs. GA vs. AA | GG + GA vs. AA | GG vs. GA + AA |
| BMI(kg/m$^2$) | 18.93±3.19 | 18.02±3.23 | 20.73±5.11 | | 0.086 | **0.037** | 0.926 |
| HGS(kg) | 16.57±4.49 | 15.99±3.98 | 18.42±4 | | 0.181 | 0.072 | 0.783 |
| SLJ(cm) | 147.44±22.82 | 135.41±17.42 | 139.38±19.99 | | 0.144 | 0.886 | 0.061 |
| Sit-ups(number) | 24.94±6.64 | 21.11±5.29 | 20.88±6.49 | | 0.076 | 0.331 | **0.023** |
| Push-ups(number) | 16.94±5.02 | 15.7±4.63 | 15.75±2.96 | | 0.613 | 0.726 | 0.321 |
| Boys | GG (28%, n = 56) | GA (47%, n = 94) | AA (25%, n = 50) | 0.40 | GG vs. GA vs. AA | GG + GA vs. AA | GG vs. GA + AA |
| BMI(kg/m$^2$) | 19.05±2.86 | 18.75±3.6 | 19.92±4.15 | | 0.732 | 0.442 | 0.855 |
| HGS(kg) | 16.08±3.49 | 17.67±3.5& | 17.93±3.5 | | 0.399 | 0.428 | 0.176 |
| SLJ(cm) | 149.42±24.78 | 144.33±18.86* | 142±18.59 | | 0.698 | 0.539 | 0.416 |
| Sit-ups(number) | 22.67±4.48# | 22.08±5.18 | 20±5.19 | | 0.434 | 0.205 | 0.395 |
| Push-ups(number) | 16.5±4.08 | 15.33±5.57 | 16.4±3.47 | | 0.788 | 0.776 | 0.673 |
| Girls | GG (26.7%, n = 40) | GA (45.3%, n = 68) | AA (28%, n = 42) | 0.25 | GG vs. GA vs. AA | GG + GA vs. AA | GG vs. GA + AA |
| BMI(kg/m$^2$) | 18.68±4.05 | 17.79±2.89 | 22.08±6.61 | | 0.091 | **0.034** | 0.969 |
| HGS(kg) | 17.55±6.33 | 14.87±3.93& | 19.23±4.97 | | 0.126 | 0.104 | 0.501 |
| SLJ(cm) | 143.5±19.79 | 128.06±12.70* | 135±23.24 | | 0.416 | 0.775 | 0.101 |
| Sit-ups(number) | 29.5±8.26# | 19.94±5.43 | 22.33±8.59 | | **0.035** | 0.864 | **0.011** |
| Push-ups(number) | 17.83±6.91 | 15.94±3.91 | 14.67±1.51 | | 0.673 | 0.368 | 0.281 |

Data are the mean ± standard deviation (SD)

HWE, Hardy-Weinberg equilibrium; distribution of the genotype expressed as frequency (percentage); # indicates a significant difference in sit-up values comparing boy and girl groups in the GG genotype, p<0.05; * indicates a significant difference in SLJ values comparing boy and girl groups in the GA genotype, p<0.05;& indicates a significant difference in HGS values comparing boy and girl groups in the GA genotype, p<0.05; p values<0.05 were considered statistically significant.

(F = 6.089, p = 0.017<0.05). Age, height, and weight were significantly associated with HGS (F = 3.255, p = 0.001<0.05; F = 3.176, p = 0.002<0.05; F = 4.186, p = 0.00<0.05), but not between sex. Height was significantly associated with push-ups (F = 2.613, p = 0.007<0.05). There was an insignificant interaction effect on SLJ between age, height, and weight. We found significant associations in boys between age and HGS, and girls between height, weight and HGS.

As shown in Table 2, in all participants or boys and girls' subgroups, the *PPARGC1A* rs8192678 (Gly482Ser) polymorphism met HWE (p > 0.05). We found that BMI in all participants and in girls was significantly associated with the dominant genetic model (p = 0.037, 0.034, respectively). In contrast, sit-ups in girls were significantly associated with the additive genetic model (p = 0.035) and the recessive genetic model (p = 0.011). In the boys' group, HGS, SLJ, sit-ups and push-ups indicators were not significantly associated with the *PPARGC1A* rs8192678 (Gly482Ser) polymorphism.

Furthermore, we compared the indicators between sexes for each genotype. In the Gly/Gly genotype there was a significant difference in sit-up values between boys and girls groups (p = 0.035). The Gly/Ser genotype showed a significant difference in HGS and SLJ values between the boy and girl groups (p = 0.048,0.014). In the Ser/Ser genotype there was a nonsignificant difference in the indicators between the boy and girl groups.

## 4. Discussion

In this study, we analyzed for the first time the association between genetic polymorphisms in the *PPARGC1A* rs8192678 (Gly482Ser) and phenotypic indicators of muscle strength in

untrained schoolchildren, and compared the variability of phenotypic indicators of muscle fitness across genotypes and genetic models.

The Ser482 allele distribution of the population in this study (MAF(minor allele frequency) = 0.49) is close to the population of Southern Han Chinese(CHS, MAF = 0.452) from the 1000 Genomes Project, and higher than the population of Han Chinese in Beijing (CHB, MAF = 0.374) and Caucasians (MAF = 0.361), indicating that the distribution of the PPARGC1A rs8192678 (Gly482Ser) polymorphism is ethnically and geographically different.

Summarizing the association between genes and skeletal muscle fibers, Ahmetov II et al. noted that in untrained individuals, the lateral thigh muscles the proportion of slow twitch (type I) fiber is approximately 50% (range 5–90%) and their conversion to fast twitch fiber is uncommon. The variability in the proportion of type I fiber observed in human muscle has a genetic component of approximately 40–50%, suggesting that the composition of muscle fiber type is determined by gene and environment [22].

PGC-1α is a mitochondrial cofactor involved in regulating glucose and lipid transport and oxidation, and therefore has an important role in skeletal muscle fiber morphogenesis and mitochondrial biogenesis [30]. The Gly482Ser has been identified as an important genetic polymorphism allele in athletic performance in athletes [31]. The Gly482Ser polymorphism in the *PPARGC1A* gene regulates muscle adaptation to training induction and is associated with cardiorespiratory fitness in athletes [28]. The frequency of the Ser482 allele is lower in white male dominant endurance athletes, whereas the Gly482 homozygote favors improved aerobic capacity [32]. The frequency of the Ser482 genotype in Polish and Russian athletes was significantly lower than that in controls, lower muscle levels affected aerobic capacity, and the Gly482 allele may be associated with endurance performance [33].

A recent meta-analysis showed that the results for the Gly482 allele were more favorable to potential sport performance than those for the Ser482 allele Bonferroni correction of data from related studies revealed that carrying the Gly482 allele had a highly significant effect on strength in Caucasian populations (p = 0.003). In contrast, it had a void effect in Asian populations [26]. In a study of aerobic training in young Han Chinese males in northern China, $VO_2$max levels were found to be higher in carriers of the Gly/Ser+Gly/Gly genotype than in the AA group, suggesting an effect of Gly482Ser on aerobic indices of $VO_2$max [34]. The Gly482 allele was positively correlated with the proportion of slow muscle fibers (r = 0.5, p = $4.0 \times 10^{-4}$) [22].

In skeletal muscle, PGC-1α plays an important role in regulating mitochondrial biogenesis and adaptation to aerobic training, regulating exercise-induced changes in muscle fibers toward a slower phenotype and protecting against muscle atrophy [22, 30, 33]. Improving the oxidative capacity of muscle fibers by increasing the number and activity of mitochondria through upregulation of nuclear respiratory factor and mitochondrial transcription factor led to increased mitochondrial DNA replication and gene transcription. Overexpression of PGC-1α increased the proportion of oxidative type I fibers, while PGC-1α knockout (KO) mice exhibited a shift from oxidative type I and IIA to IID/X and IIB fibers [35]. In this study, the sit-ups and push-ups, indicators related to muscular endurance, were lower in the girl group than in the boy group of the Gly/Ser+Ser/Ser genotypes but higher in the girl group than in the boy group of the Gly/Gly genotype. There was a significant difference between the sexes in the sit-up indicator. Studies have shown that gender-related differences in muscle development can lead to differences in physical function, as muscular strength is proportional to the cross-sectional area of the muscle and the growth curve of strength is essentially the same. It has been suggested that the magnitude of muscular endurance largely depends on the percentage of type I muscle fibers, with the percentage of type I muscle fibers in boys increasing progressively with age between 0 and 7 years and decreasing between the ages of 10 and 35 years. The

percentage of type I muscle fibers did not change much from adolescence to young adulthood in girls, but in boys the change in muscle fiber type was more pronounced in girls [36]. Further analysis revealed that the sit-up index was significantly associated with both additive and recessive models of the Gly482 allele in girls, suggesting that the Gly482 allele has a significant effect on type I fiber expression in females, with the best muscle endurance of the Gly/Gly homozygote in the girl group. In contrast, none of the three models were significantly correlated in the boy group, suggesting that the Gly482 allele may have a diminished effect on endurance-related type I muscle fibers in Chinese boys. In addition, in this study, boys were higher than girls in all three genotype groups in this study for the muscle strength-related index SLJ. There were significant gender differences in the Gly/Ser genotype. The boys were lower than the girls, except for the Gly/Ser genotype, where HGS was significantly higher than the girls.

The study showed that SLJ performance and HGS improved with age, with boys and girls aged 9–17 years improving by 5.5% and 2.6%, respectively, with corresponding improvements in HGS of 14.2% and 9.3% for boys and girls, respectively [37]. These data suggest that the related progression in maturation appears more favorable to HGS than SLJ performance. Increased fat-free muscle mass would improve SLJ performance and HGS by increasing muscle firing capacity. Increases in fat mass, on the other hand, would impair SLJ performance (through increased force requirement demands) but not HGS (a weight-independent strength task) [38]. BMI has essentially correlates with HGS. Larger body size implies longer long bones (e.g. ulna and radius), lengthening arm muscle fibers and thus contributing to HGS [39]. Although height and weight are positively correlated with HGS in adolescence, the effect of these variables is much smaller than the effect of gender or age [40]. Studies have reported that no significant differences in HGS were found between boys and girls up to the age of 10 years. From 11 years onward, boys had stronger grip and pinch strength than girls and stronger HGS in the dominant hand than in the non-dominant hand [41]. When analyzed in conjunction with BMI metrics, we found consistent results with higher HGS values for higher BMI and no sex differences. The analysis of the three genetic models with the Gly482 allele as the dominant gene showed no correlation between BMI and the Gly482 allele in boys. In contrast, BMI and the dominant genetic model of the Gly482 allele were significantly correlated in the girls' group, and the mean values of BMI and HGS in girls were Ser/Ser>Gly/Gly>Gly/Ser group. Studies have shown that those carrying the Ser482 allele have higher low-density lipoprotein (LDL) cholesterol levels, higher insulin resistance and a higher risk of developing diabetes. Myles et al. found the association between Gly482Ser genotypes and BMI in Tongans but not Maori and the 482Ser as T2D risk allele in Pacific populations [42]. Cadzow et al. found no statistically significant link between Gly482Ser and BMI in European, African 1000 Genome Project, Asian and Polynesian populations [43].

Recent studies have found that carrying the Ser482 allele affects insulin sensitivity, lipid metabolism, impaired binding of its product to coactivator of myocyte enhancer Factor 2 (MEF2), and detracts from the exercise-induced conversion of fast to slow muscle fibers [27], which could explain the enhanced effects of the Ser482 allele in this study that may promote muscle growth and development and affect BMI and type II muscle fibers in girls. However, in a study of the *PPARGC1A* polymorphism and muscle fiber type in Japanese females, found a higher percentage of type I fibers carrying the Ser/Ser genotype than type II [22].

A study of endurance, strength, speed performance in 586 untrained male Iranian adolescents in the standing long jump and 20 m fold run found that the *PPARGC1A* rs8192678 (Gly482Ser) polymorphism was associated with endurance-related phenotypes and endurance capacity [30]. In contrast, the Ser allele genotype was associated with an anaerobic threshold, slow muscle fibers [33], more mitochondrial activity and greater peak $VO_2$ after aerobic training. Another aspect is that plasmids bearing Gly482 or Ser482 of the PGC-1α protein show

that the variant is less efficient as a coactivator of myocyte enhancer Factor 2C (MEF2C). This transcription factor that regulates glucose transport in skeletal muscle [35]. In untrained muscle, exercise-induced expression of the *PPARGC1A* gene via the typical promoter may be regulated by systemic factors, whereas in trained muscle the typical promoter shows constitutive expression at rest and after exercise. Exercise-induced expression of the *PPARGC1A* gene via alternative promoters has been associated with intramuscular factors and with activation of CREB-regulated transcription coactivator 2(CRTC2)-cAMP-responsive element-binding protein 1(CREB1) [44].

## 5. Conclusion

The results showed that there were no significant differences between the sexes in height, weight and BMI in Han schoolchildren in southern China and that the HGS index was age-related in boys and height- and weight-related in girls. The Gly482 allele was hypothesized to affect on the expression of type I fiber in skeletal muscle in girls, while the Ser482 allele affected on type II fiber in girls. The two alleles had little genetic effect on boys. We supposed that there might be an association between the *PPARGC1A* rs8192678 (Gly482Ser) polymorphism and individual differences in muscle fiber type-related muscle health phenotypes in Han Chinese girls in southern China.

This polymorphism may affect lipid metabolism and muscle fiber conversion in schoolchildren's skeletal muscle. In the future, we will continue to expand the sample size, age group and index and explore the adaptation of different genotypes of Gly482Ser to exercise modulation of skeletal muscle and its effect on muscle strength and endurance. We can guide schoolchildren's exercise scientifically in a more targeted manner and promote their physical development and fitness.

### 5.1. Practical applications

1. The PPARGC1A rs8192678 (Gly482Ser) polymorphism is associated with lipid metabolism and muscle fiber conversion in schoolchildren's skeletal muscle.

2. The Gly482 and Ser482 alleles had little genetic effect on boys.

3. There might be an association between the PPARGC1A rs8192678 (Gly482Ser) polymorphism and individual differences in muscle fiber type-related phenotypes in Han Chinese girls in southern China.

4. According to the association between the PPARGC1A rs8192678 (Gly482Ser) polymorphism and muscle fitness, We can guide schoolchildren's exercise scientifically in a more targeted manner and promote their physical development and fitness.

### 5.2. Limitation and further research

In the future, we will continue to expand the sample size and age group, the index, and to explore the adaptation of different genotypes of Gly482Ser to exercise modulation of skeletal muscle and its effect on muscle strength and endurance.

### 5.3. Informed consent

Written informed consent was obtained from parents. Wuhan White Plain Technology Co. was responsible for the genotyping of participates.

## Author Contributions

**Formal analysis:** Qi Wei.

**Project administration:** Qi Wei.

**Writing – original draft:** Qi Wei.

**Writing – review & editing:** Qi Wei.

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
