## [Decision Letter · Decision Letter 0]

23 Jan 2023

PONE-D-22-35030Association between the PPARGC1A Gly482Ser polymorphism and muscle fitness in Chinese schoolchildrenPLOS ONE

Dear Dr. wei,

Thank you for submitting your manuscript to PLOS ONE. After careful consideration, we feel that it has merit but does not fully meet PLOS ONE’s publication criteria as it currently stands. Therefore, we invite you to submit a revised version of the manuscript that addresses the points raised during the review process.

ACADEMIC EDITOR:

Dear Authors,

Two experts in the field revised your current manuscript and recognised some points that should be addressed.

We look forward to receiving your revised manuscript.

Kind regards,

Javier Abián-Vicén, Ph.D.

Academic Editor

PLOS ONE

5. Please include your tables as part of your main manuscript and remove the individual files. Please note that supplementary tables (should remain/ be uploaded) as separate "supporting information" files

Reviewers' comments:

Reviewer's Responses to Questions

**Comments to the Author**

1. Is the manuscript technically sound, and do the data support the conclusions?

Reviewer #1: No

Reviewer #2: No

2. Has the statistical analysis been performed appropriately and rigorously? 

Reviewer #1: N/A

Reviewer #2: N/A

3. Have the authors made all data underlying the findings in their manuscript fully available?

Reviewer #1: Yes

Reviewer #2: No

4. Is the manuscript presented in an intelligible fashion and written in standard English?

Reviewer #1: Yes

Reviewer #2: Yes

5. Review Comments to the Author

Reviewer #1: I congratulate the authors on the study.

However, there are some points to improve.

Abstract: completely revise it. Review the background regarding muscle development in children, specify the definition and problem correlation of the PPARGC1A gene. Enter the definition first and then the gene in parentheses. Also, improve the explanation of materials and methods, are unclear. Improve the difference between Gly/Gly and Gly/Ser before explaining the results.

Improve explanation of findings and conclusion.

Introduction: Expand the background and bibliographic references regarding muscle development in children, the correlation with health and the possible negative consequences at a physiological and pathological level in case it is not corrected. Improving the binding to the gene and the respective polymorphisms and their consequences.

Improve exposure of the problem.

Insert line 86 the various types of tests used for the evaluation of physical strength, after the explanation of the problem, increase the bibliographic references on the validity of the tests.

Materials and Methods: Write a paragraph on statistical analysis and how it was evaluated. A table could be inserted with the tests carried out, the repetitions, the modalities and the respective bibliographic references.

Discussion: improve the exposition of the problem and the background as in the introduction with more in-depth analysis of the existing bibliography in the literature. Also, improve exposure regarding the innovation that the study can bring to research. Improve the link between BMI and Gly482, increase bibliographic references and exposure of the link.

Reviewer #2: The manuscript under consideration: "Association between the PPARGC1A Gly482Ser polymorphism and muscle fitness in Chinese schoolchildren" is an interesting article on an important topic in PLOS ONE.

1. The argument in the introduction is weak. The hypothesis of the study is unclear.

2. Although it discusses children and adolescents, the age range covered is 7~12 years old. It should be modified a bit more to an introduction appropriate for this study.

3. How did the authors determine the sample appropriate size?

4. Please add a methodological rationale for the assessment of physical function such as grip strength, citing previous research.

5. Please add the data, including who measured it and the years of experience.

5. The inclusion and exclusion criteria are unclear. A detailed description of how participants were selected is needed.

6. If you want to emphasize more what we have said in the discussion, it would be better to perform a multivariate analysis, such as a logistic regression analysis.

7. I believe there will be gender differences in physical function in the upper grades. Please consider conducting separate analyses for males and females.

8. What is the most newly found in this study?

6. PLOS authors have the option to publish the peer review history of their article (what does this mean?). If published, this will include your full peer review and any attached files.

Reviewer #1: **Yes: **Giovanni Messina

Reviewer #2: No

---

## [Author Response · Author response to Decision Letter 0]

10 Mar 2023

Reviewer #1: I congratulate the authors on the study.

However, there are some points to improve.

Abstract: completely revise it. Review the background regarding muscle development in children, specify the definition and problem correlation of the PPARGC1A gene. Enter the definition first and then the gene in parentheses. Also, improve the explanation of materials and methods, are unclear. Improve the difference between Gly/Gly and Gly/Ser before explaining the results.Improve explanation of findings and conclusion.

Response to reviewers: Due to the word limit of the abstract, the background of muscle development in schoolchildren is detailed in INTRODUCTION. The Gly482Ser polymorphism of the PPARGC1A gene includes two allele (Gly,Ser), which are divided into three genotypes Gly/Gly,Gly/Ser,Ser/Ser are described in the methods. Other changes have been colour coded in revision.

Introduction: Expand the background and bibliographic references regarding muscle development in children, the correlation with health and the possible negative consequences at a physiological and pathological level in case it is not corrected. Improving the binding to the gene and the respective polymorphisms and their consequences.

Improve exposure of the problem.

Response to reviewers: The background and bibliographic references haver been expanded and colour coded in revision.

Insert line 86 the various types of tests used for the evaluation of physical strength, after the explanation of the problem, increase the bibliographic references on the validity of the tests.

Response to reviewers: The bibliographic references on the validity of the tests have been in the introducion ,the references 10-17 are the latest studies and reviews assessing the validity of muscle assessment tests in children.

Materials and Methods: Write a paragraph on statistical analysis and how it was evaluated. A table could be inserted with the tests carried out, the repetitions, the modalities and the respective bibliographic references.

Response to reviewers: The statistical analysis and how it was evaluated were described in Line 149-154.The bibliographic references of the tests were marked.

Discussion: improve the exposition of the problem and the background as in the introduction with more in-depth analysis of the existing bibliography in the literature. Also, improve exposure regarding the innovation that the study can bring to research. Improve the link between BMI and Gly482, increase bibliographic references and exposure of the link.

Response to reviewers: This study was aimed to investigate the association between genetic polymorphism and muscle in children. One of the innovations was that there were no previous genetic studies of the PPARGC1A polymorphism in children's muscle, only in athletes and adults, and the second innovation was to replace the invasive muscle fiber experiment with a muscle evaluation index. As the existing literature focuses on athletes and adults, there are few studies on genetic polymorphisms in children's muscles and the depth of our analysis is limited, and we will keep exploring this direction in the future with more in-depth analysis . The link between BMI and Gly482 were improved with bibliographic references in Line 286-290.

Reviewer #2: The manuscript under consideration: "Association between the PPARGC1A Gly482Ser polymorphism and muscle fitness in Chinese schoolchildren" is an interesting article on an important topic in PLOS ONE.

1. The argument in the introduction is weak. The hypothesis of the study is unclear.

Response to reviewer:Muscle health has continued to decline in recent years in children's fitness tests, but children are unable to perform pioneering muscle type tests. Based on the studies of the PPARGC1A gene polymorphism associated with muscle in athletes and adults, it is noted that this locus polymorphism can be involved in the transformation and regulation of muscle fibres, The study was chosen to investigate the association of this locus with muscle evaluation indicators in children, with a view to finding the genetic background of muscle development in children, providing accurate fitness guidance and improving muscle health in children.

2. Although it discusses children and adolescents, the age range covered is 7~12 years old. It should be modified a bit more to an introduction appropriate for this study.

Response to reviewer: The legal age range for primary education in China is 7-12 years old, and this study is a random sample selected from primary schools, children and adolescents are modified to schoolchildren in the discussion.

3. How did the authors determine the sample appropriate size?

Response to reviewer:The test originated from a project by the General Administration of Sports of China (GAC), for part of a study on the monitoring of children's physical fitness, with a sample size defined by the GAC testing team according to geographical distribution and statistical sample requirements. This subject group undertook the sample size for this study.

4. Please add a methodological rationale for the assessment of physical function such as grip strength, citing previous research.

Response to reviewer: The HGS, SLJ, sit-ups are internationally tests and the push-ups are based on the DMT6-18. The references 10-17 are the latest studies and reviews assessing the validity of muscle assessment tests in children.

5. Please add the data, including who measured it and the years of experience.

Response to reviewer: The data were shown in table 2. 26 volunteers from the School of Physical Education at Three Gorges University who were involved in assisting with the testing and data collection. Wuhan White Plain Technology Co. was responsible for the genotyping of participates.The author of this paper has been involved in research on children's physical education testing for 18 years, supervising and training 26 volunteers, supervising and guiding test norms in the field during testing, and verifying the data.

6. The inclusion and exclusion criteria are unclear. A detailed description of how participants were selected is needed.

Response to reviewer: The subjects of this study were seleted randomly from the school students and the inclusion criteria have been detailed in the METHODS.

7. If you want to emphasize more what we have said in the discussion, it would be better to perform a multivariate analysis, such as a logistic regression analysis.

Response to reviewer: Because the dependent variables of the study were continuous variables rather than acategorical variables, multiple linear regression analysis was used in this study.

8. I believe there will be gender differences in physical function in the upper grades. Please consider conducting separate analyses for males and females.

Response to reviewer: Definitely, there were gender differences in physical function shown in Table 2. We analysed the indicators of the continuous variable against the genetic model for boys and girls by linear regression analysis.

9. What is the most newly found in this study?

Response to reviewer: This study was aimed to investigate the association between genetic polymorphism and muscle in children. One of the innovations was that there were no previous genetic studies of the PPARGC1A polymorphism in children's muscle, only in athletes and adults, and the second innovation was to replace the invasive muscle fiber experiment with a muscle evaluation index. We found an association between the PPARGC1A Gly482Ser polymorphism and individual differences in muscle fiber type-related phenotypes in Han Chinese girls in southern China.

---

## [Decision Letter · Decision Letter 1]

10 Apr 2023

Association between the PPARGC1A Gly482Ser polymorphism and muscle fitness in Chinese schoolchildren

PONE-D-22-35030R1

Dear Dr. wei,

We’re pleased to inform you that your manuscript has been judged scientifically suitable for publication and will be formally accepted for publication once it meets all outstanding technical requirements.

Kind regards,

Javier Abián-Vicén, Ph.D.

Academic Editor

PLOS ONE

Additional Editor Comments (optional):

Congratulations for your work!

Reviewers' comments:

Reviewer's Responses to Questions

**Comments to the Author**

1. If the authors have adequately addressed your comments raised in a previous round of review and you feel that this manuscript is now acceptable for publication, you may indicate that here to bypass the “Comments to the Author” section, enter your conflict of interest statement in the “Confidential to Editor” section, and submit your "Accept" recommendation.

Reviewer #2: All comments have been addressed

Reviewer #3: All comments have been addressed

2. Is the manuscript technically sound, and do the data support the conclusions?

Reviewer #2: Yes

Reviewer #3: Yes

3. Has the statistical analysis been performed appropriately and rigorously? 

Reviewer #2: Yes

Reviewer #3: Yes

4. Have the authors made all data underlying the findings in their manuscript fully available?

Reviewer #2: No

Reviewer #3: Yes

5. Is the manuscript presented in an intelligible fashion and written in standard English?

Reviewer #2: Yes

Reviewer #3: Yes

6. Review Comments to the Author

Reviewer #2: Thank you for giving me the opportunity to peer review your valuable paper. The authors carefully addressed all of the reviewer comments. I have no further comments.

Reviewer #3: Congratulations for your paper. I think the document meets the criteria established by Plos One to be published.

7. PLOS authors have the option to publish the peer review history of their article (what does this mean?). If published, this will include your full peer review and any attached files.

Reviewer #2: No

Reviewer #3: No
